# CITB: A Benchmark for Continual Instruction Tuning

**Zihan Zhang[1], Meng Fang[2], Ling Chen[1], Mohammad-Reza Namazi-Rad[3]**

[1]University of Technology Sydney  [2]University of Liverpool
[3]University of Wollongong

Zihan.Zhang-5@student.uts.edu.au, Meng.Fang@liverpool.ac.uk
Ling.Chen@uts.edu.au, mrad@uow.edu.au

## Abstract

Continual learning (CL) is a paradigm that aims to replicate the human ability to learn and accumulate knowledge continually without forgetting previous knowledge and transferring it to new tasks. Recent instruction tuning (IT) involves fine-tuning models to make them more adaptable to solving NLP tasks in general. However, it is still uncertain how instruction tuning works in the context of CL tasks. This challenging yet practical problem is formulated as Continual Instruction Tuning (CIT). In this work, we establish a CIT benchmark consisting of learning and evaluation protocols. We curate two long dialogue task streams of different types, **InstrDialog** and **InstrDialog++**, to study various CL methods systematically. Our experiments show that existing CL methods do not effectively leverage the rich natural language instructions, and fine-tuning an instruction-tuned model sequentially can yield similar or better results. We further explore different aspects that might affect the learning of CIT. We hope this benchmark will facilitate more research in this direction[1].

## 1 Introduction

Recent studies have shown that multi-task instruction tuning (IT) makes language models better zero-shot learners (Wei et al., 2022; Sanh et al., 2022; Wang et al., 2022; Chung et al., 2022; Longpre et al., 2023). IT fine-tunes pre-trained language models (PLMs) on various tasks with natural language instructions (Fig.1) and can achieve remarkably well generalization to unseen tasks.

Despite their impressive performance, these instruction-tuned PLMs still fall short on domain-specific tasks due to the limited exposure to relevant knowledge and vocabulary from the training corpus (Luo et al., 2023). Moreover, PLMs are static after deployment, and there is no mechanism

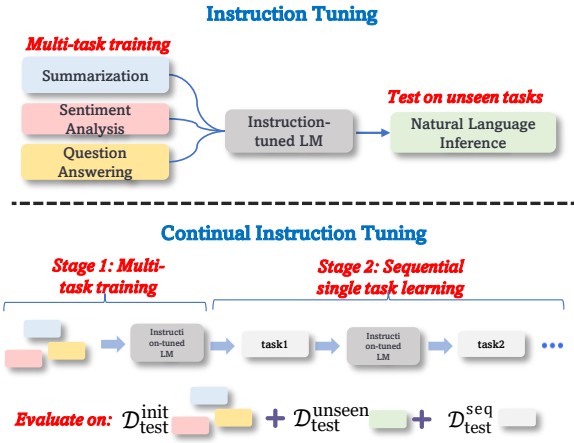

Figure 1: Illustration of proposed continual instruction tuning (CIT). Unlike previous works, we evaluate the instruction-tuned model on the initial training, unseen, and newly learned tasks.

to update themselves or adapt to a changing environment (Zhang et al., 2023; Bubeck et al., 2023).

Continual learning (CL) aims to enable information systems to learn from a continuous data stream across time (Biesialska et al., 2020). Therefore, it is promising to leverage CL for instruction-tuned PLMs to continually adapt to new domains and tasks without costly re-training. Despite its importance, it is non-trivial to alleviate *catastrophic forgetting*, a phenomenon in which previously learned knowledge or abilities are degraded due to overwritten parameters (McCloskey and Cohen, 1989). Moreover, enabling knowledge transfer is also essential since many tasks are similar and have common knowledge (Ke et al., 2021).

Unfortunately, there is little work on applying CL for IT and has only been explored in rather specific settings. Scialom et al. (2022) continually fine-tune a T0 (Sanh et al., 2022) on eight new tasks with memory replay to avoid forgetting. Despite effectiveness, they need to store large number of instances per task in memory, which is too costly when scaling to a larger number of tasks.

---

[1]Code and data are available at https://github.com/hyintell/CITB.

In addition, they do not study knowledge transfer between tasks. Yin et al. (2022) propose to use history task instructions to reduce forgetting and enable knowledge transfer. However, they do not compare with commonly adopted CL methods, which makes the effectiveness of other CL methods unknown. Moreover, they only evaluate the model on the newly learned tasks while largely ignoring previously learned tasks during the multi-task training stage (Fig.1). They also overlook the intrinsic ability of the instruction-tuned model on unseen tasks. Lastly, both of them use different evaluation metrics and setups, which creates an obstacle to comparing different techniques and hinders the development of this field.

To this end, we first formulate this practical yet under-explored problem as **Continual Instruction Tuning** (CIT). Then, we propose a first-ever benchmark suite to study CIT systematically. Our benchmark, CITB, consists of both learning and evaluation protocol and is built on top of the recently proposed SuperNI dataset (Wang et al., 2022). We create two CIT task streams: **InstrDialog** stream, which consists of 19 dialogue-related tasks spanning three categories; **InstrDialog++** stream, which includes all the tasks in **InstrDialog** stream and 19 additional tasks selected from broad categories and domains. Using the two long task streams, we implement various CL methods to study forgetting and knowledge transfer under the setup of CIT. We find that directly fine-tuning an instruction-tuned model sequentially yields competitive performance with existing CL methods. With further investigation, we find that rich natural language instructions enable knowledge transfer and reduce forgetting, which is barely fully leveraged by current CL methods. We conduct comprehensive experiments to explore what effects the learning of CIT. We hope our CITB benchmark will serve as a helpful starting point and encourage substantial progress and future work by the community in this practical setting. To summarize, our main contributions are:

- We formulate the problem of CIT and establish a benchmark suite consisting of learning and evaluation protocols.
- We curate two long task streams of various types based on the SuperNI dataset to study different setups of CIT.
- We implement various CL methods of different categories, conduct extensive experiments

and ablation studies to analyze the lack of current practices, and propose a future direction.

## 2 Related Work

**Instruction Tuning.** Much effort has been made recently to use natural language instructions to solve multiple tasks concurrently or to align with human preferences (Touvron et al., 2023; Zhou et al., 2023; OpenAI, 2023). Unlike simple and short prompts (Liu et al., 2021), natural language instructions (Fig.2) can be more comprehensive, including components such as task definition, in-context examples (Brown et al., 2020), and explanations. Through IT, PLMs learn to complete tasks by following instructions, which enables them to solve *new* tasks by following instructions without learning (i.e., generalization ability). Ideally, we expect the instruction-tuned model to understand any given task instruction so that an end user can directly leverage the model to solve the task without the need to annotate a large dataset and train it. Unfortunately, despite the instruction-tuned models such as FLAN (Wei et al., 2022; Longpre et al., 2023), T0 (Sanh et al., 2022), and Tk-Instruct (Wang et al., 2022) showing strong generalization performance to their evaluation tasks, there is still a sizeable gap compared with supervised training, which limits the usage of the models. From a practical point of view, a desirable instruction-tuned model should be able to extend its ability by *continually* learning those under-performed tasks or any new task, while not forgetting the old ones.

**Continual Learning.** In contrast to multi-task learning, continually fine-tuning a model on tasks might lead to *catastrophic forgetting* (McCloskey and Cohen, 1989), where the model forgets previously acquired knowledge after learning new tasks. In CL literature, approaches to overcoming catastrophic forgetting can be grouped into three categories (Biesialska et al., 2020; Ke and Liu, 2023). *Regularization*-based methods use an additional loss to prevent important parameters of previous tasks from being updated (Kirkpatrick et al., 2017; De Lange et al., 2019). *Replay*-based methods store and replay a small subset of training data from previous tasks to prevent forgetting (Rebuffi et al., 2017; Scialom et al., 2022); *Architecture*-based methods introduce task-specific components for new tasks and isolate parameters of old tasks (Madotto et al., 2021; Zhu et al., 2022). However, the effectiveness of these CL methods for

**Definition:**
In this task, you are given a sentence and a question, you would be asked to create the answer which is contained in the sentence provided.

**Positive Example 1**
**Input:** *"Sentence: Heat from the sun causes the most evaporation of water from a lake. Question: Which of these causes the MOST evaporation of water from a lake?"*
**Output:** *"Heat from the Sun"*
**Explanation:** *"The output is correct as ... it as the 'Heat from the sun'"*

**Negative Example 1**
**Input:** *"Sentence: Gas has no definite volume and no definite shape. Question: Which state of matter has no definite volume and no definite shape?"*
**Output:** *"Matter"*
**Explanation:** *"The correct answer to ... hence it is incorrect."*

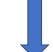

**Input:** *"Sentence: The lens of the eye is a(n) convex shape. Question: What shape is the lens of the eye?"*
**Expected Output:** "convex"

Figure 2: An example of natural language instruction that consists of a descriptive task definition, one positive and one negative in-context example with explanation (Wang et al., 2022). Given a task instruction and the input of a test instance, a model needs to produce the desired output.

CIT remains unknown. CIT differs from traditional CL in heavily relying on comprehensive instructions. Can previous methods fully leverage the rich instructions to avoid forgetting while facilitating knowledge transfer between tasks? Moreover, many proposed CL methods target tasks of specific types (e.g., text classification, relation extraction) (Huang et al., 2021; Qin and Joty, 2022; Zhu et al., 2022), while CIT can learn broad tasks of different categories because of the natural language instructions[2]. Yin et al. (2022); Scialom et al. (2022); Mok et al. (2023) study a similar problem using IT in the CL setting, but no benchmarks are built for different categories of CL methods. To tackle CIT, it is essential to establish an unified benchmark to compare existing approaches and promote the development of this field. However, to our best knowledge, CIT is still immature and no public benchmark is available.

## 3 Preliminaries

**Instruction Tuning (IT).** Following previous studies (Wei et al., 2022; Sanh et al., 2022; Wang et al., 2022), each task $t \in \mathcal{T}$ consists of its natural language instruction $I^t$ and a set of $N$ input-output

---

[2]All tasks can be filled into a natural language instruction template and transformed into a text-to-text format (Wei et al., 2022).

instances $\mathcal{D}^t = \left\{ \left( x_i^t, y_i^t \right) \in \mathcal{X}^t \times \mathcal{Y}^t \right\}_{i=1}^N$, which can be split into the training $\mathcal{D}_{\text{train}}^t$, validation $\mathcal{D}_{\text{dev}}^t$ and test sets $\mathcal{D}_{\text{test}}^t$. Each instance is filled into an instruction template such that different tasks can be transformed into a unified text-to-text format (Fig.2). IT aims to learn a model $f : I^t \times \mathcal{X}^t \to \mathcal{Y}^t$ that can predict the output $y_i^t$ given the task instruction $I^t$ and an input $x_i^t$. In general, the model is first trained on a mixture of tasks ($\mathcal{T}_{\text{seen}}$) and then evaluated for its zero-shot generalization ability on held-out tasks ($\mathcal{T}_{\text{unseen}}$), where $\mathcal{T}_{\text{seen}} \cap \mathcal{T}_{\text{unseen}} = \emptyset$. The model is expected to learn to follow instructions via the training tasks and then solve new tasks with only the help of task instructions.

## 4 Continual Instruction Tuning Benchmark

In this section, we first formalize the CIT problem (Fig.1). Then, we present the learning and evaluation protocol of our framework CITB. Lastly, we describe the data for creating the benchmark.

### 4.1 Continual Instruction Tuning

In contrast to static IT, which only learns a fixed set of tasks ($\mathcal{T}_{\text{seen}}$), the model should be able to keep learning new tasks without catastrophically forgetting previously learned knowledge and facilitate knowledge transfer if possible. Let us expand the definition such that we have a set of $T$ tasks $\mathcal{T}_{\text{seq}} = \{t_1, \cdots, t_T\}$ that arrives sequentially. Note that the tasks in the stream can be any type and are not restricted to specific categories or domains. Similarly, each task $t_j \in \mathcal{T}_{\text{seq}}$ has a natural language instruction $I^{t_j}$, training $\mathcal{D}_{\text{train}}^{t_j}$, validation $\mathcal{D}_{\text{dev}}^{t_j}$ and test sets $\mathcal{D}_{\text{test}}^{t_j}$. Likewise traditional CL, the goal of CIT is to learn a *single* model $f$ from $\mathcal{T}_{\text{seq}}$ sequentially.

**CIT vs. Traditional CL.** While sharing similar desiderata with traditional CL, CIT differs in that: (1) it pays more attention to effectively leveraging the rich natural language instructions to prevent catastrophic forgetting and encourage knowledge transfer; (2) because of the multi-task nature of instructions, all tasks can be formatted in the unified text-to-text format, therefore CIT can learn any task or domain instead of a few specific tasks or domains; (3) after learning a few tasks, the model should have learned how to follow instructions to complete tasks, therefore, we expect fewer training instances required and higher knowledge transfer for future tasks.

## 4.2 Learning Protocol of CIT Benchmark

A non-instruction-tuned model (e.g., T5; Raffel et al. 2020) may struggle to understand instructions if trained only on a task sequentially. It is also against our motivation to extend a model's ability that is already instruction-tuned. Therefore, we separate the learning process into two stages.

**Stage 1: Initial Multi-task Fine-tuning.** To teach a model a better understanding of task instructions, we first fine-tune the model on instruction data. Suppose we have another group of $M$ tasks $\mathcal{T}_{\text{init}} = \{t_1, \cdots, t_M\}$ that also equips with natural language instructions, where $\mathcal{T}_{\text{init}} \cap \mathcal{T}_{\text{seq}} = \emptyset$. We fine-tune a base pre-trained model on the training set (i.e., $\mathcal{D}_{\text{train}}^{\text{init}} = \bigcup_{i=1}^{M} \mathcal{D}_{\text{train}}^{t_i}$) of the mixed $M$ tasks to get an instruction-tuned model, denoted as $f_{\text{init}}$. After training, most of the training data $\mathcal{D}_{\text{train}}^{\text{init}}$ is unavailable for subsequent sequential learning, but a memory $\mathcal{M}_{\text{init}}$ ($\mathcal{M}_{\text{init}} \ll |\mathcal{D}_{\text{train}}^{\text{init}}|$) that stores a small portion of training instances is accessible. We use this model as the starting point to conduct the subsequent learning.

**Stage 2: Sequential Single Task Fine-tuning.** To keep extending knowledge of the instruction-tuned $f_{\text{init}}$, we fine-tune it on the training set $\mathcal{D}_{\text{train}}^{t_j}$ of each task $t_j$ in the stream $\mathcal{T}_{\text{seq}}$. Similarly, when learning the task $t_j$, the training data of previous tasks in the stream (i.e., $\mathcal{D}_{\text{train}}^{\text{seq}} = \bigcup_{i=1}^{j-1} \mathcal{D}_{\text{train}}^{t_i}$, $i < j < T$) is unavailable, but a small memory $\mathcal{M}_{\text{seq}}$ can be used for training.

## 4.3 Evaluation Protocol of CIT Benchmark

**Evaluation Process.** After learning each task $t_j$ ($1 < j < T$) in stream $\mathcal{T}_{\text{seq}}$, we consider three datasets to measure the model's performance. (1) Similar to the standard CL, we evaluate the model on the test sets of all previously learned tasks in the stream, the test set of the current task, and the test set of the next task, denoted as $\mathcal{D}_{\text{test}}^{\text{seq}}$. This helps us measure whether the model forgets previous knowledge and whether it is helpful to learn future tasks; (2) We evaluate the model on the test sets of the $M$ tasks that are used in stage 1 to teach the model how to follow instructions, denoted as $\mathcal{D}_{\text{test}}^{\text{init}} = \bigcup_{i=1}^{M} \mathcal{D}_{\text{test}}^{t_i}$. This is where different from the conventional CL. In CL, previous works only evaluate downstream tasks in the stream but not the tasks during the pre-training phase because such data is generally not accessible to end-users (Ke et al., 2022). (3) Since multi-task instruction-tuned

models have shown strong zero-shot generalization to unseen tasks (Wang et al., 2022), our initial model trained in stage 1 might also have zero-shot generalization to some unseen tasks $\mathcal{T}_{\text{unseen}}$, where $\mathcal{T}_{\text{init}} \cap \mathcal{T}_{\text{seq}} \cap \mathcal{T}_{\text{unseen}} = \emptyset$. Let $\mathcal{D}_{\text{test}}^{\text{unseen}}$ be the test sets of all tasks in $\mathcal{T}_{\text{unseen}}$. We evaluate the model on $\mathcal{D}_{\text{test}}^{\text{unseen}}$ if it is available. To sum up, once a new task is learned, the model will be evaluated on:

$$D = \mathcal{D}_{\text{test}}^{\text{seq}} + \mathcal{D}_{\text{test}}^{\text{init}} + \mathcal{D}_{\text{test}}^{\text{unseen}} \qquad (1)$$

In CIT, it is more critical for the instruction-tuned model to maintain its existing abilities than learn new ones because it can solve multiple tasks by following instructions. Otherwise, if it forgets many tasks, there is no point in using such a model than a task-specific one. Therefore, it is essential to evaluate on $\mathcal{D}_{\text{test}}^{\text{init}}$ and $\mathcal{D}_{\text{test}}^{\text{unseen}}$.

**Evaluation Metrics.** Due to the diversity of the tasks in CIT and the open-ended generation nature of the text-to-text format, we follow Wang et al. (2022) to use *ROUGE-L* (Lin, 2004) to measure the aggregated performance of each task. They have shown that *ROUGE-L* generally works well for both generation and classification tasks.

Following Lopez-Paz and Ranzato (2017) and Biesialska et al. (2020), we also use CL-related metrics to measure the learning procedure. Let $a_{j,i}$ be the *ROUGE-L* score of the model on the test set of task $t_i$ right after training on task $t_j$, we define the following:

**Average *ROUGE-L*** (AR), which measures the average performance of the model on all tasks after the final task $t_T$ is learned:

$$\mathbf{AR}_T = \frac{1}{T} \sum_{i=1}^{T} a_{T,i} \qquad (2)$$

We use **Final *ROUGE-L*** (FR) to measure the performance of the model on $\mathcal{D}_{\text{test}}^{\text{init}}$ and $\mathcal{D}_{\text{test}}^{\text{unseen}}$, respectively, after the final task $t_T$ is learned.

**Forward Transfer** (FWT), which measures how much the model can help to learn the new task. FWT also tests the model's zero-shot generalization to new tasks:

$$\mathbf{FWT}_T = \frac{1}{T-1} \sum_{i=2}^{T} a_{i-1,i} \qquad (3)$$

**Backward Transfer** (BWT), which measures the impact that continually learning on subsequent tasks has on previous tasks:

$$\mathbf{BWT}_T = \frac{1}{T-1} \sum_{i=1}^{T-1} (a_{T,i} - a_{i,i}) \qquad (4)$$

Notably, positive BWT indicates that subsequent tasks can improve the performance of previous tasks, while negative value implies knowledge forgetting.

## 4.4 Data Curation

In this work, we adopt the recently proposed SuperNI (Wang et al., 2022) dataset to establish the benchmark. SuperNI consists of more than 1,600 NLP tasks, spanning a diverse variety of 76 broad task types, such as language generation, classification, question answering, and translation. Moreover, each task is equipped with an instruction and a set of instances, and all the instances can be transformed into the text-to-text format. Therefore, the dataset is suitable for studying CIT. The official training set of SuperNI[3] consists of 756 English tasks spanning 60 broad NLP categories, while 119 tasks from 12 categories are used for zero-shot evaluation. We keep the official 119 evaluation tasks untouched and create two CIT task streams from the 756 training tasks.

**InstrDialog Stream.** Dialogue is an important field to study continual learning because new tasks, domains or intents are continuously emerging (Madotto et al., 2021). To investigate how a model learns new dialogue data under the setup of CIT, we carefully curate *all* dialogue-related tasks from the training set of SuperNI to form the CIT task stream. Specifically, we use 4 tasks from dialogue state tracking, 11 tasks from dialogue generation, and 4 tasks from intent identification, resulting in a total of 19 dialogue tasks, i.e., $|\mathcal{T}_{\text{seq}}| = 19$. We remove tasks that are excluded by the official task splits[4].

**InstrDialog++ Stream.** Because of the multi-task nature of instructions, an instruction-tuned model can learn any new task with different types (§4.1). To study how different types of tasks and how a long-task curriculum affects CIT, we first include all 19 dialogue tasks from the **InstrDialog** stream, then we manually select the other 19 tasks

from the remaining training task set. We intentionally select tasks from broad categories, including sentence ordering, style transfer, toxic language detection, and others. In total, we have 38 tasks of 18 categories (3 categories from InstrDialog and 15 categories from the new 19 tasks), i.e., $|\mathcal{T}_{\text{seq}}| = 38$.

The remaining training tasks can be used for stage 1 initial multi-task fine-tuning (§4.2). In summary, the number of initial fine-tuning tasks available is $M = |\mathcal{T}_{\text{init}}| = 718$, and we use the official 119 test task sets as $\mathcal{T}_{\text{unseen}}$ to evaluate whether the performance deteriorates for unseen tasks after learning new tasks. For all tasks, we fill instances in a natural language instruction template and transform them into a unified text-to-text format (§3). Unless otherwise specified, we use the instruction template consisting of the task definition and two positive examples for all tasks because it generally yields the best performance (Wang et al., 2022). See an example of natural language instructions in Fig.2, and the selected tasks in Table 5. We study the effect of the instruction template in §6.3.

## 5 Experiments

Using our CITB benchmark, we conduct experiments on various popular CL methods of different kinds. We describe our experiment setups and compared methods in this section.

### 5.1 Setup

**Model.** We use the LM-adapted version of T5-small (Raffel et al., 2020), which is further trained with a language modeling objective. We initialize a T5 model from HuggingFace[5]. Since it is costly to fine-tune on all 718 tasks, we randomly select 100 tasks from $\mathcal{T}_{\text{init}}$ and fine-tune T5 to obtain an instruction-tuned model $f_{\text{init}}$ (§4.2), which has learned to understand some instructions and can act as a good starting point to conduct subsequent learning. Note that the 100 randomly selected training tasks do not overlap with **InstrDialog** and **InstrDialog++**, but their task categories might overlap with the categories in **InstrDialog++**.

**Train/Dev/Test Splits.** Since the number of instances in each task is imbalanced and a large number of training instances do not help generalization in IT (Wang et al., 2022), we follow Wang et al. (2022) use a fixed size of 500/50/100 instances per task as the train/dev/test set for the **InstrDialog**

---

[3]https://github.com/allenai/natural-instructions

[4]https://github.com/allenai/natural-instructions/tree/master/splits

[5]https://huggingface.co/google/t5-small-lm-adapt

| Method | InstrDialog | | | $\mathcal{T}_{\text{init}}$ FR | $\mathcal{T}_{\text{unseen}}$ FR | Mem. | +P (Tun) | Time |
|---|---|---|---|---|---|---|---|---|
| | AR | FWT | BWT | | | | | |
| FT-no-init | $29.6_{2.1}$ | $8.0_{0.2}$ | $-10.8_{2.3}$ | $17.3_{0.5}{}^{\dagger}$ | $16.5_{1.3}{}^{\dagger}$ | 0 | 0 (1) | $0.5_{0.02}$ |
| AdapterCL | $8.1_{0.1}$ | $9.4_{0.7}$ | $-21.9_{0.9}$ | - | - | 0 | T*0.02 (0.02) | $\mathbf{0.4}_{0.03}$ |
| Init | $22.5^{\dagger}$ | - | - | 43.5 | $\mathbf{36.5}^{\dagger}$ | - | - | - |
| FT-init | $35.7_{0.2}$ | $18.5_{0.7}$ | $-4.6_{0.2}$ | $38.6_{0.3}$ | $32.3_{0.6}{}^{\dagger}$ | 0 | 0 (1) | $0.6_{0.01}$ |
| L2 | $35.6_{0.1}$ | $17.5_{0.5}$ | $-3.8_{1.2}$ | $39.4_{0.4}$ | $34.9_{1.2}{}^{\dagger}$ | 0 | 1 (1) | $0.6_{0.1}$ |
| EWC | $34.5_{0.6}$ | $16.8_{0.4}$ | $-6.8_{1.5}$ | $37.0_{0.1}$ | $32.5_{0.5}{}^{\dagger}$ | 0 | 2 (1) | $1.1_{0.2}$ |
| AGEM (10) | $33.2_{0.4}$ | $19.1_{0.1}$ | $-7.3_{1.0}$ | $38.6_{1.1}$ | $32.4_{0.0}{}^{\dagger}$ | (T+M)*10 | 0 (1) | $1.1_{0.1}$ |
| AGEM (50) | $34.9_{0.9}$ | $18.1_{1.0}$ | $-6.0_{0.9}$ | $37.7_{0.1}$ | $32.6_{1.0}{}^{\dagger}$ | (T+M)*50 | 0 (1) | $1.3_{0.1}$ |
| Replay (10) | $38.4_{0.7}$ | $\mathbf{23.7}_{0.0}$ | $-1.3_{0.5}$ | $42.7_{0.7}$ | $32.4_{0.4}{}^{\dagger}$ | (T+M)*10 | 0 (1) | $1.4_{0.04}$ |
| Replay (50) | $40.4_{0.0}$ | $22.9_{0.1}$ | $\mathbf{1.6}_{1.2}$ | $\mathbf{47.1}_{0.5}$ | $31.8_{1.0}{}^{\dagger}$ | (T+M)*50 | 0 (1) | $3.2_{0.5}$ |
| Multi | $\mathbf{42.1}_{0.6}$ | - | - | $44.7_{1.3}$ | $32.8_{0.9}{}^{\dagger}$ | 0 | 0 (1) | $1.1_{0.2}$ |

Table 1: Performance of different methods on the **InstrDialog** stream. Means and standard deviations are reported. † means zero-shot performance. "Mem." means the number of instances stored in the memory for each task; $T$ is the total number of tasks in the stream and $M$ is the number of tasks used for initial training. "+P" means the percentage of additional parameters added for each task, measured by the total parameters of the base model; "Tun" is the portion of tunable parameters during training. "Time" is the average hours for each method to complete the task stream. Best numbers are in bold.

stream. For **InstrDialog++**, since it has a longer task sequence, to save computational cost, we use 100/50/100 instances per task instead. For $\mathcal{T}_{\text{init}}$, we use 100/50/100 instances per task and 100 instances per task for $\mathcal{T}_{\text{unseen}}$. We study the effect of different numbers of training instances in §6.3.

## 5.2 Baselines and Compared Methods

We implement commonly used CL methods from three categories (Biesialska et al., 2020) to benchmark CIT.

*Regularization-based* methods rely on a fixed model capacity with an additional loss term to consolidate previously gained knowledge while learning subsequent tasks. We use **L2** and **EWC** (Kirkpatrick et al., 2017), which uses a fisher information matrix to reduce forgetting by regularizing the loss to penalize the changes made to important parameters of previous tasks.

*Replay-based* methods store a small subset of training instances from previous tasks in a memory. The data are replayed later to reduce forgetting. We adopt **Replay**, which saves random instances from each task in a memory and then jointly trains the model on new task data and the old data in the memory; **AGEM** (Chaudhry et al., 2019), which adds constraint to prevent parameter update from increasing the loss of each previous task. The loss of previous tasks is calculated using the instances stored in the memory.

*Architectural-based* methods introduce task-specific parameters to the base model to prevent subsequent tasks from interfering with previously learned parameters. We adopt **AdapterCL** (Madotto et al., 2021), which freezes the pretrained model and trains a residual Adapter (Houlsby et al., 2019) for each task independently.

Apart from the three categories of CL, we also implement *instruction-based* baselines because all previous CL methods are not designed for CIT. No prior work had tried to fine-tune an instruction-tuned model sequentially without any mechanism for preventing forgetting or encouraging knowledge transfer. However, it is commonly considered a performance lower bound in CL literature. To this end, we propose to continually fine-tune the initial instruction-tuned model (§5.1) on subsequent tasks, named as **FT-init**. As a direct comparison, we initialize a new T5 model, which is not tuned on any instruction data, and we continually fine-tune it (**FT-no-init**). In addition, we also report the performance of the initial instruction-tuned model (**Init**), which is the starting point before subsequent learning. Lastly, we jointly fine-tune a T5 model using all the data, including the training data used in stage 1 and the training data of all subsequent tasks in the stream (**Multi**). This is often regarded as the performance upper bound in CL and does not have catastrophic forgetting and knowledge transfer.

| Method | InstrDialog++ | | | $\mathcal{T}_{\text{init}}$ | $\mathcal{T}_{\text{unseen}}$ |
| | AR | FWT | BWT | FR | FR |
|---|---|---|---|---|---|
| FT-no-init | $31.3_{2.4}$ | $26.8_{0.3}$ | $-5.2_{1.4}$ | $28.9_{0.6}^{\dagger}$ | $31.2_{2.1}^{\dagger}$ |
| AdapterCL | $20.7_{0.2}$ | $14.3_{0.9}$ | $-7.3_{1.1}$ | - | - |
| Init | $30.5^{\dagger}$ | - | - | $43.5$ | $\mathbf{36.5}^{\dagger}$ |
| FT-init | $34.8_{0.3}$ | $\mathbf{29.8}_{0.5}$ | $-\mathbf{2.8}_{0.6}$ | $40.8_{0.2}$ | $35.9_{0.5}^{\dagger}$ |
| L2 | $36.1_{0.2}$ | $27.9_{0.3}$ | $-4.0_{1.2}$ | $41.1_{0.4}$ | $34.9_{1.1}^{\dagger}$ |
| EWC | $38.6_{0.4}$ | $28.6_{0.5}$ | $-4.0_{1.8}$ | $41.3_{0.3}$ | $34.0_{0.7}^{\dagger}$ |
| AGEM (10) | $39.3_{0.6}$ | $28.9_{0.2}$ | $-3.8_{1.1}$ | $40.3_{0.9}$ | $34.1_{0.3}^{\dagger}$ |
| Replay (10) | $43.1_{0.7}$ | $29.0_{0.4}$ | $-3.6_{0.8}$ | $44.0_{0.5}$ | $30.1_{0.6}^{\dagger}$ |
| Multi | $\mathbf{44.9}_{0.5}$ | - | - | $\mathbf{44.6}_{0.9}$ | $33.9_{0.5}$ |

Table 2: Performance of different methods on the **InstrDialog++** stream. $\dagger$ means zero-shot performance.

## 5.3 Implementation Details

For both **InstrDialog** stream and **InstrDialog++** stream, we conduct continual instruction tuning using the same initial instruction-tuned model for all methods except **FT-no-init**, **AdapterCL**, and **Multi**. For these methods, we initialize a new T5 model. For **AGEM** and **Replay**, we experiment on a memory size of 10 and 50, i.e., we set the memory size $\mathcal{M}_{\text{init}}$ to 10 and 50, same as $\mathcal{M}_{\text{seq}}$. We jointly train the data in $\mathcal{M}_{\text{init}}$, $\mathcal{M}_{\text{seq}}$, and the new task data for these two methods. For **AdapterCL**, we use a bottleneck of 100. Due to limited computing resource, we randomly permute the task streams and run all experiments using three random seeds. We refer this as task order 1. We study the effect of task orders in §6.3. The selected tasks for task stream **InstrDialog** and **InstrDialog++** are listed in Table 5. The task orders are listed in Table 6. More details are in Appendix A.

## 6 Results and Analysis

In this section, we report the performance of various baselines discussed in §5.2 on our benchmark.

### 6.1 Results on InstrDialog Stream

Table 1 shows each method's overall performance and resource requirement after continually learning the **InstrDialog** stream. We have the following observations:

**First**, all methods except AdapterCL have improved AR, compared to the zero-shot performance (22.5) of the starting point model Init. This shows CIT can extend a model's knowledge. In contrast, although AdapterCL is parameter-efficient and does not rely on memory, it performs even

worse than Init. We conjecture that AdapterCL fails to learn instructions effectively because it is initialized from a non-instruction-tuned model (T5) and the few tunable parameters restrict it from learning complex instructions.

**Second**, among all baselines, Replay generally have the best performance. All methods except Replay (50) have negative BWT, meaning that they all suffer from catastrophic forgetting. Furthermore, forgetting on $\mathcal{T}_{\text{init}}$ and $\mathcal{T}_{\text{unseen}}$ is even worse, which demonstrates that the ability of the initial instruction-tuned model Init has deteriorated after learning new dialogue tasks. We also find storing more examples in memory improves Replay but does not significantly help AGEM. It might be because the constraints added to the loss are roughly the same, no matter how many instances are stored. Despite used additional parameters, regularization-based L2 and EWC perform similar to other baselines. Multi overall performs well, with the highest AR and improved FR on $\mathcal{T}_{\text{init}}$, however, it also forgets tasks in $\mathcal{T}_{\text{unseen}}$. Replay (50) has a higher FR on $\mathcal{T}_{\text{init}}$ than Multi because the 5,000 instances stored in $\mathcal{M}_{\text{init}}$ are jointly trained multiple times when learning subsequent tasks (§5.3), leading to better data fitting.

**Third**, FT-init performs surprisingly well on all metrics and is competitive to L2, EWC, and AGEM. This finding contradicts the common sense in CL that simply fine-tuning a model sequentially would lead to catastrophic forgetting because all parameters can be freely updated in learning new tasks (Kirkpatrick et al., 2017). Even for FT-no-init, which is not tuned on any instruction data, shows increased AR after learning the 19 dialogue tasks. This raises a question: do various CL methods truly

mitigate forgetting and promote knowledge transfer in CIT? We hypothesize that the rich natural language instructions lead to the remarkable performance of the baselines (§6.3).

## 6.2 Results on InstrDialog++ Stream

The performance of all methods after learning the **InstrDialog++** steam is shown in Table 2. We observe most of the same findings as in §6.1, except that:

**First**, Init has a higher (30.5 vs. 22.5) zero-shot performance on this long stream than on **InstrDialog**, as in Table 1. We analyze that the categories of the selected 100 training tasks (§5.1) overlap with the categories in the stream, which enables more knowledge transfer of tasks between the same category because of the similar natural language instructions. For example, both sets have tasks from sentiment analysis and toxic language detection. In contrast, Init did not learn dialogue tasks, thus showing lower generalization on **InstrDialog**.

**Second**, we can see improved performance for almost all methods compared to Table 1, especially on $\mathcal{T}_{\text{init}}$ and $\mathcal{T}_{\text{unseen}}$. For FT-init and FT-no-init, the improvements of FWT and BWT are particularly significant, reaching the best among all CL methods.

Combining the results on the two streams from Table 1 and 2, we find that catastrophic forgetting exists in CIT. However, learning a longer task stream and diverse tasks of different types leads to better knowledge transfer and lower forgetting.

| Instr. | FT-init | | | FT-no-init | | |
| --- | --- | --- | --- | --- | --- | --- |
| | AR | FWT | BWT | AR | FWT | BWT |
| id | 13.9 | 10.0 | -30.3 | 11.5 | 8.0 | -27.7 |
| def | 38.0 | 21.7 | -6.9 | 33.8 | 16.1 | -7.3 |
| def+2p | 34.8 | 29.8 | -2.8 | 31.3 | 26.8 | -5.2 |
| def+2p +2n | 40.4 | 28.1 | -7.1 | 33.8 | 22.3 | -7.3 |

Table 3: Effect of instruction templates on **InstrDialog++**. "id": only use the short task id; "def": only use descriptive task definitions; "def+2p": use task definition and two positive examples; "def+2p+2n": use additional two negative examples.

## 6.3 Ablation Studies

In this section, we investigate the reason why instruction-based baselines (FT-init and FT-no-init) perform as well as or even better than conventional

| Instr. | FT-init | | FT-no-init | |
| --- | --- | --- | --- | --- |
| | $\mathcal{T}_{\text{init}}$ | $\mathcal{T}_{\text{unseen}}$ | $\mathcal{T}_{\text{init}}$ | $\mathcal{T}_{\text{unseen}}$ |
| id | 3.6 | 2.8 | 1.0 | 1.7 |
| def | 25.8 | 23.6 | 20.3 | 22.4 |
| def+2p | 40.8 | 35.9 | 28.9 | 31.2 |
| def+2p+2n | 37.7 | 35.7 | 27.2 | 33.1 |

Table 4: Effect of instruction templates on $\mathcal{T}_{\text{init}}$ and $\mathcal{T}_{\text{unseen}}$.

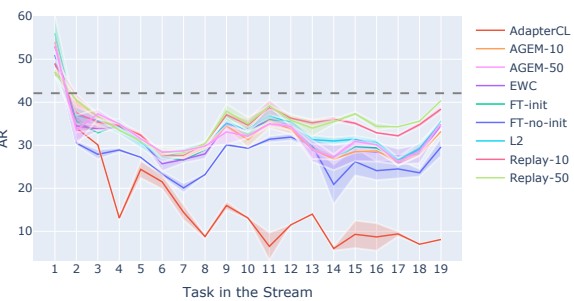

Figure 3: AR of each method during learning the **InstrDialog** stream (task order 1).

CL methods (§6.1 & §6.2). We also explore different aspects that might affect CIT.

**Rich instructions enable knowledge transfer and reduce forgetting in CIT.** We use the same setup as in Table 2, except for using different instruction templates. Results in Table 3 confirms our hypothesis that the remarkable performance of FT-init and FT-no-init comes from the rich natural language instructions. When only using the task Id (e.g., task565_circa_answer_generation) as instruction without descriptive task definition or in-context examples, simply fine-tuning the model sequentially yields low AR, FWT, and BWT, which aligns with the findings in CL literature. Additionally, providing in-context examples (2p or 2p+2n) generally improves performance. However, although task performance (AR) is improved with two additional negative examples, we witness decreased knowledge transfer (FWT and BWT). For the model that is not fine-tuned on any instruction data (FT-no-init), we find it worse than FT-init, showing the benefits of the initial multi-task training on instruction data.

Similar observations are found on $\mathcal{T}_{\text{init}}$ and $\mathcal{T}_{\text{unseen}}$ in Table 4, where the model catastrophically forgets its initial abilities after learning a long stream of tasks. Providing descriptive task definitions significantly boosts task performance as well

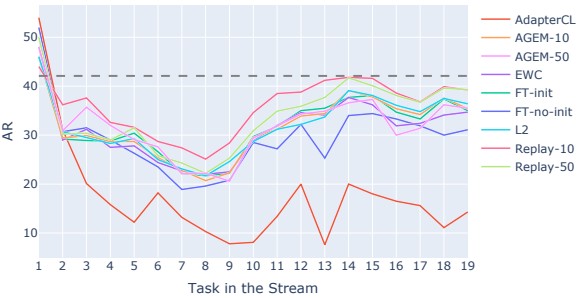

Figure 4: AR of each method during learning the **Instr-Dialog** stream (task order 2).

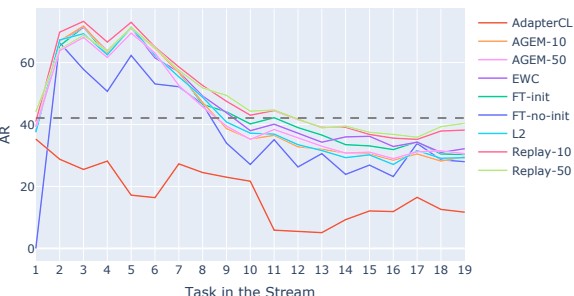

Figure 5: AR of each method during learning the **Instr-Dialog** stream (task order 3).

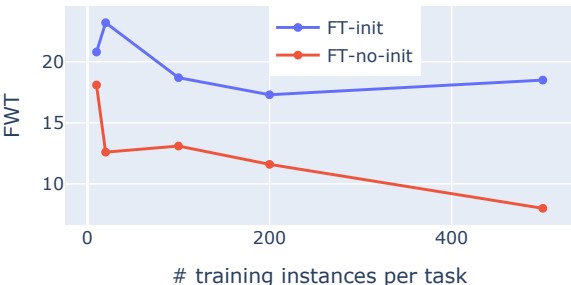

Figure 6: Effect of training instances per task on FWT.

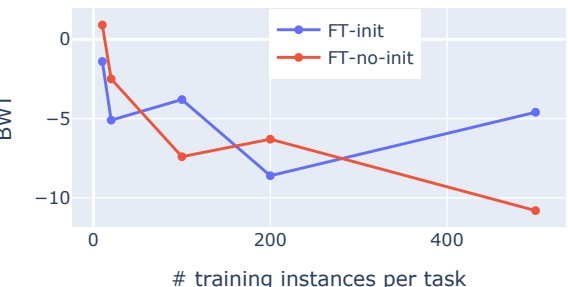

Figure 7: Effect of training instances per task on BWT.

as facilitates knowledge transfer. Moreover, it also maintains the model's generalization ability on unseen tasks. Combining the results in Table 1, 2, 3, and 4, we find that those conventional CL methods do not fully leverage the instructions to reduce forgetting and facilitate knowledge transfer while learning continuously because the naive FT-init and FT-no-init can also achieve the same. This calls for novel CL methods designed for CIT.

**Task types and learning order matter to CIT.** To explore how task types and orders in a stream affects CIT, we randomly permute the **InstrDialog** stream to get two new task orders and conduct the same learning as Table 1. We present the intermediate learning trends of all 19 tasks in Fig.3, Fig.4 and Fig.5. One can see from the plots that all baselines are highly affected by task orders, fluctuating dramatically when different tasks are learned first. We argue that it is because the task difficulties and similarities vary a lot. Learned tasks transfer knowledge through the instructions to new tasks of the same type, therefore facilitating its learning. For example, the last task in order 1 (Fig.3) is a type of dialogue generation, which is the dominant task type in the stream (11/19, §4.4), therefore all baselines are improved. However, all baselines reach below Multi after learning all 19 tasks, demonstrat-

ing knowledge forgetting will eventually appear if learning longer enough tasks.

**A large number of training instances do not help knowledge transfer.** We vary the number of instances per task used for learning the **InstrDialog** stream from [10, 20, 100, 200, 500]. As shown in Fig.6 and Fig.7, FWT and BWT gradually decrease when the number of training instances is scaled to large values. It aligns with the findings by Wang et al. (2022) in standard IT, where large number of instances do not help generalization to unseen tasks, we also find it is true in CIT. Additionally, we find instruction-tuned models (FT-init) have better generalization to new tasks (FWT) than the model not fine-tuned on any instruction data (FT-no-init). This shows that, after learning a few tasks, the model have learned how to follow instructions to complete tasks, thus fewer training instances are required for new tasks.

## 7 Conclusion

In this work, we establish a benchmark for continual instruction tuning, with two 19 and 38 long task streams to be learned sequentially. We implement and compare various continual learning methods of different types using the benchmark to study their effectiveness under this new domain. We conduct extensive ablation studies to analyze the lack of current practices, and propose a future direction.

## Limitations

We identify our limitations as follows. First, due to limited resources, all experiments in this work use T5-small (LM-adapted) as the backbone, which might not entirely reflect continual instruction tuning in general. As Wang et al. (2022) points out, there is a sizable gap between the smaller models and the 11B or 3B models in generalizing to new tasks. Second, when creating the two CIT task streams, we only use English tasks from the SuperNI dataset (Wang et al., 2022). In future, it can be extended to multilingual task streams to study cross-language continual instruction tuning. Third, we follow the SuperNI dataset to use ROUGE-L as an aggregated metric to evaluate all tasks. Although it acts as a good proxy for the model's overall performance, it might not serve as an effective measurement for some specific tasks. Fourth, while we selected diverse tasks to form the InstrDialog and InstrDialog++ task streams, we did not analyse the characteristics of these tasks (Kim et al., 2023). In future, we consider to select better source tasks and study how source tasks affect CIT.

## Acknowledgements

This work is supported by TPG Telecom. We would like to thank anonymous reviewers for their valuable comments.

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

## A   Implementation Details

For both **InstrDialog** stream and **InstrDialog++** stream, we conduct continual instruction tuning using the same initial instruction-tuned model for all methods except **FT-no-init**, **AdapterCL**, and **Multi**. For these methods, we initialize a new T5 model. Since AdapterCL needs to train a task-specific adapter for each task, it is too costly to train the 100 adapters for it, therefore we initialize it from a T5 model. For AdapterCL, we use a bottleneck of 100.

The initial instruction-tuned model (§5.1) is trained on 100 tasks with 100 instances per task (in total 10,000). We use a maximum of 15 epochs with learning rate 1e-05. We perform checkpoint selection using the development set.

For **AGEM** and **Replay**, we experiment on a memory size of 10 and 50, i.e., we set the memory size $\mathcal{M}_{\text{init}}$ to 10 and 50, same as $\mathcal{M}_{\text{seq}}$. We jointly train the data in $\mathcal{M}_{\text{init}}$, $\mathcal{M}_{\text{seq}}$, and the new task data for these two methods.

For L2 and EWC, we tune the regularization term from $[0.001, 0.01, 0.1]$, and use 0.01. For AdapterCL, we use learning rate of 1e-3. For other methods, we use 1e-5. For the **InstrDialog** stream, we use a batch size of 4 for EWC and 8 for all other methods; for the **InstrDialog++** stream, we use a batch size of 16. For all experiments except AdapterCL, we train a maximum epoch of 15 and perform early stopping with 3 patience to avoid overfitting; for AdapterCL, we use a larger epoch of 20 with patience of 5.

The selected tasks for task stream **InstrDialog** and **InstrDialog++** are listed in Table 5. The task orders are listed in Table 6. All experiments are done in an RTX3090 Ti with 24GB VRAM.

| Task Category | Number | Task Name | Domain |
|---|---|---|---|
| **Dialogue Generation** | 1 | task565_circa_answer_generation | Dialogue |
| | 2 | task574_air_dialogue_sentence_generation | Dialogue |
| | 3 | task576_curiosity_dialogs_answer_generation | Dialogue, Commonsense |
| | 4 | task611_mutual_multi_turn_dialogue | Dialogue |
| | 5 | task639_multi_woz_user_utterance_generation | Dialogue |
| | 6 | task1590_diplomacy_text_generation | Dialogue, Game |
| | 7 | task1600_smcalflow_sentence_generation | Dialogue, Commonsense |
| | 8 | task1603_smcalflow_sentence_generation | Dialogue |
| | 9 | task1714_convai3_sentence_generation | Dialogue |
| | 10 | task1729_personachat_generate_next | Dialogue |
| | 11 | task1730_personachat_choose_next | Dialogue |
| **Intent Identification** | 12 | task294_storycommonsense _motiv_text_generation | Story |
| | 13 | task573_air_dialogue_classification | Dialogue |
| | 14 | task848_pubmedqa_classification | Medicine |
| | 15 | task1713_convai3_sentence_generation | Dialogue |
| **Dialogue State Tracking** | 16 | task766_craigslist_bargains_classification | Dialogue |
| | 17 | task1384_deal_or_no_dialog_classification | Dialogue, Commonsense |
| | 18 | task1500_dstc3_classification | Dialogue, Public Places |
| | 19 | task1501_dstc3_answer_generation | Dialogue, Public Places |
| **Style Transfer** | 20 | task927_yelp_negative_to_positive_style_transfer | Reviews |
| **Sentence Ordering** | 21 | task1549_wiqa_answer_generation_missing_step | Natural Science |
| **Word Semantics** | 22 | task459_matres_static_classification | News |
| **Text Categorization** | 23 | task379_agnews_topic_classification | News |
| **Pos Tagging** | 24 | task347_hybridqa_incorrect_answer_generation | Wikipedia |
| **Fill in The Blank** | 25 | task1360_numer_sense_multiple_choice_qa_generation | Commonsense |
| **Program Execution** | 26 | task1151_swap_max_min | Mathematics |
| | 27 | task636_extract_and_sort_unique_alphabets_in_a_list | Mathematics |
| **Question Generation** | 28 | task301_record_question_generation | News |
| | 29 | task082_babi_t1_single_supporting _fact_question_generation | Commonsense |
| **Misc.** | 30 | task306_jeopardy_answer_generation_double | Knowledge Base |
| | 31 | task1427_country_region_in_world | Countries |
| **Coherence Classification** | 32 | task298_storycloze_correct_end_classification | Story |
| **Question Answering** | 33 | task864_asdiv_singleop_question_answering | Mathematics |
| | 34 | task598_cuad_answer_generation | Law |
| **Summarization** | 35 | task1553_cnn_dailymail_summarization | News |
| **Commonsense Classification** | 36 | task1203_atomic_classification_xreact | Commonsense |
| **Wrong Candidate Generation** | 37 | task967_ruletaker_incorrect_fact _generation_based_on_given_paragraph | Commonsense |
| **Toxic Language Detection** | 38 | task1607_ethos_text_classification | Social |

Table 5: List of tasks selected from SuperNI (Wang et al., 2022)

| Task Order | Task's IDs in order |
|---|---|
| Order1 | 848 611 565 1714 574 1590 1730 294 576 1600 1500 639 1729 1501 1713 766 1603 1384 573 |
| Order2 | 848 1603 1714 565 611 1590 1600 639 294 1500 1384 1713 1501 576 574 1729 766 573 1730 |
| Order3 | 1713 576 1384 294 573 611 1729 1600 574 1590 848 639 766 1501 565 1603 1730 1500 1714 |

Table 6: Task orders for three runs of the InstrDialog Stream.