# OpenReview forum: "CITB: A Benchmark for Continual Instruction Tuning"
_EMNLP/2023/Conference — EMNLP 2023 Findings_

### Official Review · Reviewer_eWi8 · 2023-08-01

**Typos Grammar Style And Presentation Improvements:** See reasons to reject.
**Soundness:** 3

**Excitement:**

3: Ambivalent: It has merits (e.g., it reports state-of-the-art results, the idea is nice), but there are key weaknesses (e.g., it describes incremental work), and it can significantly benefit from another round of revision. However, I won't object to accepting it if my co-reviewers champion it.

**Paper Topic And Main Contributions:**

The paper provides a continual instruction tuning (CIT) benchmark. The paper provides two long task streams for instruction tuning and systematically studies the existing continual learning methods.

**Reasons To Accept:**

* The paper proposed a new continual instruction tuning (CIT) benchmark that consists of dialogue tasks.
* The paper provides adequate baseline results of CL methods on the CIT problem and shows that instruction tuning is a great way to provide forward/backward transfers.
* No obvious concerns regarding to the benchmark construction.

**Reasons To Reject:**

* As the framework is looking at instruction tuning, generative replay methods may be a great addition to be included in the benchmark.
* The paper has no mention of code that will be provided or open-sourced, nor supplementary materials provided for the code. Unfortunately, without code, the contribution as a benchmark is degraded significantly.

**Reproducibility:**

4: Could mostly reproduce the results, but there may be some variation because of sample variance or minor variations in their interpretation of the protocol or method.

**Reviewer Confidence:**

3: Pretty sure, but there's a chance I missed something. Although I have a good feel for this area in general, I did not carefully check the paper's details, e.g., the math, experimental design, or novelty.

---

> ### Author Rebuttal · Authors · 2023-08-29
>
> Dear Reviewer eWi8,
>
> We appreciate your valuable time, review and suggestions. We want to address your concerns in detail as follows.
>
> > Q1. As the framework is looking at instruction tuning, generative replay methods may be a great addition to be included in the benchmark.
>
> **A1:** Thanks for your suggestion. Generative replay methods generate pseudo samples instead of storing true samples in the memory, which is beneficial when raw data is unavailable. We will consider exploring generative replay methods as a future work.
>
> > Q2. The paper has no mention of code that will be provided or open-sourced, nor supplementary materials provided for the code. Unfortunately, without code, the contribution as a benchmark is degraded significantly.
>
> **A2:** Thanks for your interest. As a benchmarking paper, we will release all data, code and models.

---

### Official Review · Reviewer_y4HB · 2023-08-01

**Soundness:** 4
**Typos Grammar Style And Presentation Improvements:** 1. Line 059

**Excitement:**

4: Strong: This paper deepens the understanding of some phenomenon or lowers the barriers to an existing research direction.

**Missing References:**

Please refer to my comments in "Reasons To Reject".

**Paper Topic And Main Contributions:**

The paper formulates the Continual Instruction Tuning (CIT) problem which studies how instruction tuning works in the context of continual learning (CL) tasks. A first-ever CIT benchmark is established which includes two long task streams of different types to study various CL methods systematically. The benchmark results show that existing CL methods do not effectively leverage the rich natural language instructions. Also, strikingly, unlike other CL scenarios, simply fine-tuning the model sequentially doesn't trigger a serious forgetting problem. The authors attribute this to the rich semantic information provided by the task instructions.

**Questions For The Authors:**

1. In Section 5.2, why don't you include ConTinTin which you also mentioned in the Related Works? I think it's the closest work to this paper and its method can be used in the CIT problem. If applicable, showing its results on your benchmark will be great.

**Reasons To Accept:**

1. Strictly formulating the CIT problem and providing the corresponding benchmark is very beneficial to the community. This paper did a solid work on this side by exploiting the SuperNI dataset and designing comprehensive metrics (i.e., the model is evaluated on the initial training, unseen, and newly learned tasks to reveal different aspects in CIT).
2. One interesting finding in this paper is that unlike other CL scenarios, the forgetting is not a major issue in CIT when we directly update the model sequentially. The authors attribute this to rich instructions and cleverly verify this via an ablation experiment.
3. Overall, the paper is nicely written and the experiments are extensive for a benchmarking paper.

**Reasons To Reject:**

I don't see a major point to reject this paper. However, I think the paper can be greatly improved if there could be in-depth analysis of why/how rich instructions can enable knowledge transfer and reduce forgetting in CIT. This is a very important question for the study of CIT as this is where CIT and other CL scenarios behave differently. While this question may be quite difficult to answer, a possible way to approach it may be tracking how the PLM itself change in the process of CIT. For example, [1] tracks PLM's representation ability in the class-incremental learning process and finds that the generation objective is beneficial to continual learning by preserving PLM's representation ability in the CL process. A similar tracking approach may be applied in this work; or you may investigate the changing trajectory of other internal properties of PLM in the CIT process.

[1] Class-Incremental Learning based on Label Generation, Shao et al., ACL 2023.

**Reproducibility:**

4: Could mostly reproduce the results, but there may be some variation because of sample variance or minor variations in their interpretation of the protocol or method.

**Reviewer Confidence:**

5: Positive that my evaluation is correct. I read the paper very carefully and I am very familiar with related work.

---

> ### Author Rebuttal · Authors · 2023-08-29
>
> Dear Reviewer y4HB,
>
> We appreciate your valuable time, review and suggestions. We want to address your concerns in detail as follows.
>
> > Q1. I don't see a major point to reject this paper. However, I think the paper can be greatly improved if there could be in-depth analysis of why/how rich instructions can enable knowledge transfer and reduce forgetting in CIT. This is a very important question for the study of CIT as this is where CIT and other CL scenarios behave differently. While this question may be quite difficult to answer, a possible way to approach it may be tracking how the PLM itself change in the process of CIT. For example, [1] tracks PLM's representation ability in the class-incremental learning process and finds that the generation objective is beneficial to continual learning by preserving PLM's representation ability in the CL process. A similar tracking approach may be applied in this work; or you may investigate the changing trajectory of other internal properties of PLM in the CIT process.
>
> **A1:** Thanks for your suggestion. We will provide more investigations and explanations of how rich instructions can enable knowledge transfer and reduce forgetting in CIT. Apart from your suggestion to use [1] to track the PLM's representation ability in the learning process, we also plan to use [2] to investigate task correlations and similarities. Our InstrDialog stream consists of all dialogue-related tasks, while the InstrDialog++ stream provides more diverse task categories. Using these two CL task streams and the above tools, we plan to provide more explanations and analysis.
>
> > Q2. In Section 5.2, why don't you include ConTinTin which you also mentioned in the Related Works? I think it's the closest work to this paper and its method can be used in the CIT problem. If applicable, showing its results on your benchmark will be great.
>
> **A2:** Since ConTinTin did not release their code, we tried to re-implement their method according to their paper (L446). However, we cannot reproduce the results as reported in their paper. Therefore, we did not use ConTinTin in the benchmark for a fair comparison.
>
>
> [1] Class-Incremental Learning based on Label Generation, Shao et al., ACL 2023.\
> [2] TaskWeb: Selecting Better Source Tasks for Multi-task NLP,  arXiv 2023.

---

### Official Review · Reviewer_Nb3H · 2023-08-12

**Soundness:** 3

**Excitement:**

3: Ambivalent: It has merits (e.g., it reports state-of-the-art results, the idea is nice), but there are key weaknesses (e.g., it describes incremental work), and it can significantly benefit from another round of revision. However, I won't object to accepting it if my co-reviewers champion it.

**Paper Topic And Main Contributions:**

This paper formulate and establish a benchmark for continual instruction tuning (CIT) problem. The benchmark contains two task sequences constructed from SuperNI dataset: (i) InstrDialog consisting of 19 dialog tasks and (ii) InstrDialog++ of 38 tasks from broader categories. The authors also propose evaluation protocol to assess the performance on both old, upcoming and unseen tasks. The experiments and analysis on several well-studied CL methods reveal an unexpected effectiveness of the simple fine-tuning method over other CL methods.

**Questions For The Authors:**

1. What is the practical scenario of Continual Instruction Tuning?
2. In Table 1 and Table 2, the Init model outperforms the Multi model on unseen test set. I would expect these two models have comparable performance, especially when the init training set is much larger than the sequence training set. Could the authors provide some insights on this?
3. For replay method, does the memory have fixed size? Or is each task allocated fixed number of samples to store in memory, i.e. the memory size grows as tasks arrive?

**Reasons To Accept:**

- The authors design a thoughtful evaluation protocol to evaluate on both test set of the pretrained and finetuned phase as well as unseen test set.
- Authors conduct rigorous experiments to study the behaviour of different CL methods in the CIT setting.

**Reasons To Reject:**

- One of main contribution of the paper is the CITB benchmark. I would expect analysis on task characteristics such as the instruction style, negative and positive ratio and similarity among tasks within and across init, seq and unseen test set. For example, in the same task category (e.g. dialogue generation), is the instruction similar among task? Do tasks mostly differ in the domain (covariate shift) or labels (label shift)? This can help to draw insight on the unexpected effectiveness of finetuning and the intriguing case where the Init model outperforms the Multi model on unseen test set.
- There should be a metric to account for how the learning of new task affects the performance on the init and unseen test set after training each task in the sequence. Currently, the paper only report the performance after training the final task.
- Table 2 is missing a multi-task baseline (Multi-Init) on seq training set where the model initialized with Init model.
- As stated in the limitation, the paper only evaluates on T5-small model. It's challenging to confidently assert that the current observations are applicable for larger model.

**Reproducibility:**

3: Could reproduce the results with some difficulty. The settings of parameters are underspecified or subjectively determined; the training/evaluation data are not widely available.

**Reviewer Confidence:**

3: Pretty sure, but there's a chance I missed something. Although I have a good feel for this area in general, I did not carefully check the paper's details, e.g., the math, experimental design, or novelty.

---

> ### Author Rebuttal · Authors · 2023-08-29
>
> Dear Reviewer Nb3H,
>
> We appreciate your valuable time, review and suggestions. We want to address your concerns in detail as follows.
>
> > Q1. One of main contribution of the paper is the CITB benchmark. I would expect analysis on task characteristics such as the instruction style, negative and positive ratio and similarity among tasks within and across init, seq and unseen test set. For example, in the same task category (e.g. dialogue generation), is the instruction similar among task? Do tasks mostly differ in the domain (covariate shift) or labels (label shift)? This can help to draw insight on the unexpected effectiveness of finetuning and the intriguing case where the Init model outperforms the Multi model on unseen test set.
>
> **A1:** Thanks for your suggestion. We have briefly provided the characteristics and statistics of our benchmark tasks in Appendix Fig.5 and Table 5.
> For all experiments, by default, we use the instruction template consisting of the task definition, two positive examples and no negative examples (L378) because it has been proven to be the most effective by SuperNI.
> Having said that, we will consider adding more statistics and analysis of our benchmark.
>
> > Q2. There should be a metric to account for how the learning of new task affects the performance on the init and unseen test set after training each task in the sequence. Currently, the paper only report the performance after training the final task.
>
> **A2:** This might be a misunderstanding of our evaluation protocol. As discussed in Section 4.3 L283, we evaluate the model on the **D_seq**, **D_init**, and **D_unseen** test sets **after each task is learnt**. Table 1 and Table 2 only show the final results after the last task is learnt due to space limitations.
> We have shown the intermediate learning performance (after each task) in Figures 2, 6, 7.
> However, we acknowledge that we do not have intermediate performance on the init and unseen tasks while training each task in sequence (similar to Figures 2, 6, 7). We will add these plots when we can modify our paper.
>
> > Q3. Table 2 is missing a multi-task baseline (Multi-Init) on seq training set where the model initialized with Init model.
>
> **A3:**
> There might be some misconceptions here. The **Init** model in **Table 2** is initialised from the fresh T5 model checkpoint before learning the InstrDialog++ stream (not the same mode in Table 1).
>
> Please correct us if we misunderstood your question. If we understand correctly, initialising the model from the **Init** model in **Table 1** when learning the InstrDialog++ stream (Multi-Init) does not make sense. The InstrDialog++ stream already contains all 19 tasks in the InstrDialog stream (with another 19 non-dialogue-related tasks, L361). Therefore, it does not make sense to initialise the model from the Table 1 Init model, which has already learned the 19 dialogue tasks.
>
> > Q4. As stated in the limitation, the paper only evaluates on T5-small model. It's challenging to confidently assert that the current observations are applicable for larger model.
>
> **A4:**  It is a very good suggestion and thanks for your understanding. We tried T5-large as a larger model on the InstrDialog task stream and found consistent results with T5-small. We would like to borrow more computing resources to use larger models to add more comparisons. We plan to experiment using the T5-3B, Pythia-1.4B and Pythia-2.8B and will add ablation studies on the effect of model size.
>
> > Q5. What is the practical scenario of Continual Instruction Tuning?
>
> **A5:** In practice, pre-trained models such as T5 and Llama are versatile in many tasks but might still be poor enough to follow human instructions to perform tasks.
> To improve the LM’s instruction-following ability, a common approach is to fine-tune the LM on instruction data. For instance, fine-tune T5 on dialogue/chat data so that they are more chat-friendly and can perform dialogue-related tasks. However, existing efforts only fine-tune the LM on one specific domain. To obtain an instruction-tuned LM on other domains, they need to start from the initial pre-trained LM and repeat the fine-tuning process, leading to **multiple copies of the instruction-tuned model** (such as T5-chat, T5-law). In Continual Instruction Tuning, we only have **one model** and continually improve the model on multiple domains.
>
> > Q6. In Table 1 and Table 2, the Init model outperforms the Multi model on unseen test set. I would expect these two models have comparable performance, especially when the init training set is much larger than the sequence training set. Could the authors provide some insights on this?
>
> **A6:** We use the official 119 test tasks from the SuperNI dataset as the unseen task (L372) and randomly select 100 tasks from the non-overlapping tasks to train the Init model (L395). For the Multi model, the model is trained jointly on the 100 tasks and the InstrDialog or the InstrDialog++ tasks.
>
> We conjure the reason why the Init model outperforms the Multi model on the 119 unseen tasks is that the Multi model might be overfitting and thus perform worse. We have checked the 119 unseen tasks. Although all tasks are not seen, their task categories might overlap with those in the 100 initial training tasks and InstrDialog and InstrDialog++.  For example, there are 10 dialogue-related tasks in the 119 unseen tasks. Since our base model is relatively small, training on more related data might hurt its generalization ability on unseen data and thus perform worse than the Init model.
>
> > Q7. For replay method, does the memory have fixed size? Or is each task allocated fixed number of samples to store in memory, i.e. the memory size grows as tasks arrive?
>
> **A7:** For the replay method, the memory size grows as tasks arrive. However, we experimented with two settings – we set the memory size as 10 and 50 respectively for each task, both in the initial multi-task training and sequential single-task training stage. We found that using the small replay size can already perform relatively well.

---

### Official Review · Reviewer_okNW · 2023-08-13

**Soundness:** 3

**Excitement:**

3: Ambivalent: It has merits (e.g., it reports state-of-the-art results, the idea is nice), but there are key weaknesses (e.g., it describes incremental work), and it can significantly benefit from another round of revision. However, I won't object to accepting it if my co-reviewers champion it.

**Missing References:**

[1] Kim, J., Asai, A., Ilharco, G., & Hajishirzi, H. (2023). TaskWeb: Selecting Better Source Tasks for Multi-task NLP. arXiv preprint arXiv:2305.13256.
[2] Jang, J., Kim, S., Ye, S., Kim, D., Logeswaran, L., Lee, M., ... & Seo, M. (2023). Exploring the benefits of training expert language models over instruction tuning. arXiv preprint arXiv:2302.03202.

**Paper Topic And Main Contributions:**

The authors test different continual learning methods on their proposing benchmark, CITB. CITB is consisted of InstrDialog and InstrDialog++ while using SuperNI as the seen task group. While there were a number of works studying continual learning of instruction-tuned LMs, this work organized a well established benchmark to extensively study continual learning. The findings are surprising in that CL methods do not mitigate catastrophic forgetting in the context of instruction tuning.

**Questions For The Authors:**

Q1) Was there any heuristic of choosing new tasks included in the InstrDialog++ Stream, or was it randomly chosen?
Q3) One reason why there wasn't much knowledge transfer might be due to the low task correlation between the chosen stream tasks & the remaining tasks. Additional analysis using TaskWeb[1] might be helpful to take a look into this.
Q4) Expert LM[2] is another considerable baseline, where a single expert LM is trained in order to mitigate catastrophic forgetting while preserving good unseen task performance when retrieved properly. It would be great if it is added to the baseline.

**Reasons To Accept:**

1. CITB is a well established benchmark that could further be used in future research.
2. Extensive experiments are conducted to show that previous CL methods are not very effective in more challenging settings (InstrDialog++) opposed to the argument from previous papers such as ContinualT0.

**Reasons To Reject:**

One of the reasons why CL methods do not work might be due to the small model size used in the experiments. Although it is understandable that it might have been due to limited resources, it is still questionable if the same result will hold for larger models as well.

**Reproducibility:**

4: Could mostly reproduce the results, but there may be some variation because of sample variance or minor variations in their interpretation of the protocol or method.

**Reviewer Confidence:**

4: Quite sure. I tried to check the important points carefully. It's unlikely, though conceivable, that I missed something that should affect my ratings.

---

> ### Author Rebuttal · Authors · 2023-08-29
>
> Dear Reviewer okNW,
>
> We appreciate your valuable time, review and suggestions. We want to address your concerns in detail as follows.
>
> > Q1. One of the reasons why CL methods do not work might be due to the small model size used in the experiments. Although it is understandable that it might have been due to limited resources, it is still questionable if the same result will hold for larger models as well.
>
> **A1:** It is a very good suggestion and thanks for your understanding. We tried T5-large as a larger model on the InstrDialog task stream and found consistent results with T5-small. We would like to borrow more computing resources to use larger models to add more comparisons. We plan to experiment using the T5-3B, Pythia-1.4B and Pythia-2.8B and will add ablation studies on the effect of model size.
>
> > Q2. Was there any heuristic of choosing new tasks included in the InstrDialog++ Stream, or was it randomly chosen?
>
> **A2:** We intended to explore the effect of various task categories. Based on such a strategy, we first select task categories essentially different from dialogues by nature (such as sentence ordering and style transfer) to distinguish with the InstrDialog stream. Then, from these chosen categories, we randomly select tasks.
>
> > Q3. One reason why there wasn't much knowledge transfer might be due to the low task correlation between the chosen stream tasks & the remaining tasks. Additional analysis using TaskWeb[1] might be helpful to take a look into this.
>
> **A3:** Thanks for your suggestion.
> We also note that all tasks in the InstrDialog stream are relatively similar to other task categories (i.e., have a relatively higher correlation than non-dialogue-related tasks). However, the knowledge transfer is still not very high (Table 1). We will consider using TaskWeb to analyse the impact of different tasks.
>
> > Q4. Expert LM[2] is another considerable baseline, where a single expert LM is trained in order to mitigate catastrophic forgetting while preserving good unseen task performance when retrieved properly. It would be great if it is added to the baseline.
>
> **A4:** Thanks for your suggestion.
> However, technically speaking, training a separate expert LM per task does not precisely conform to the continual learning (CL) setting because there has to be only a single model for all sequentially learned tasks in CL [1]. We will consider adding this baseline to our benchmarks for completeness.
>
>
> [1] Continual Lifelong Learning in Natural Language Processing: A Survey, COLING 2020

---

### Meta-Review · Area_Chair_3B8D · 2023-09-16

**Recommendation:** 2

**Metareview:**

The paper titled "Continual Instruction Tuning: A Benchmark Study" presents a benchmark for the continual instruction tuning (CIT) problem. The benchmark consists of two long task streams constructed from the SuperNI dataset, and the authors propose an evaluation protocol to assess performance on old, upcoming, and unseen tasks. The experiments and analysis on several well-studied continual learning (CL) methods reveal an unexpected effectiveness of the simple fine-tuning method over other CL methods.
Overall, the reviewers think that the authors design a useful evaluation protocol to evaluate on both the test set of the pretrained and finetuned stages as well as the unseen test set. However, the paper lacks detailed analysis on task characteristics such as the instruction style, negative and positive ratio, and it only evaluates on a small model, and it is challenging to confidently assert that the current observations are applicable for larger models.

---

### Decision · Program_Chairs · 2023-10-07

**Decision:**

Accept-Findings

**Comment:**

The paper titled "Continual Instruction Tuning: A Benchmark Study" presents a benchmark for the continual instruction tuning (CIT) problem. The benchmark consists of two long task streams constructed from the SuperNI dataset, and the authors propose an evaluation protocol to assess performance on old, upcoming, and unseen tasks. The experiments and analysis on several well-studied continual learning (CL) methods reveal an unexpected effectiveness of the simple fine-tuning method over other CL methods.
Overall, the reviewers think that the authors design a useful evaluation protocol to evaluate on both the test set of the pretrained and finetuned stages as well as the unseen test set. However, the paper lacks detailed analysis on task characteristics such as the instruction style, negative and positive ratio, and it only evaluates on a small model, and it is challenging to confidently assert that the current observations are applicable for larger models.